# dGPredictor: Automated fragmentation method for metabolic reaction free energy prediction and *de novo* pathway design

**Lin Wang**☯, **Vikas Upadhyay**[ID]☯, **Costas D. Maranas**[ID]*

Department of Chemical Engineering, Pennsylvania State University, University Park, Pennsylvania, United States America

☯ These authors contributed equally to this work.
* costas@psu.edu

**Data Availability Statement:** All relevant data are within the manuscript and its Supporting Information files.

## Abstract

Group contribution (GC) methods are conventionally used in thermodynamics analysis of metabolic pathways to estimate the standard Gibbs energy change ($\Delta_r G'^o$) of enzymatic reactions from limited experimental measurements. However, these methods are limited by their dependence on manually curated groups and inability to capture stereochemical information, leading to low reaction coverage. Herein, we introduce an automated molecular fingerprint-based thermodynamic analysis tool called dGPredictor that enables the consideration of stereochemistry within metabolite structures and thus increases reaction coverage. dGPredictor has comparable prediction accuracy compared to existing GC methods and can capture Gibbs energy changes for isomerase and transferase reactions, which exhibit no overall group changes. We also demonstrate dGPredictor's ability to predict the Gibbs energy change for novel reactions and seamless integration within *de novo* metabolic pathway design tools such as novoStoic for safeguarding against the inclusion of reaction steps with infeasible directionalities. To facilitate easy access to dGPredictor, we developed a graphical user interface to predict the standard Gibbs energy change for reactions at various pH and ionic strengths. The tool allows customized user input of known metabolites as KEGG IDs and novel metabolites as InChI strings (https://github.com/maranasgroup/dGPredictor).

## Author summary

The standard Gibbs energy change is commonly used to check for the feasibility of enzyme-catalyzed reactions as thermodynamics plays a crucial role in pathway design for biochemical synthesis. The group contribution methods using expert-defined functional groups have been extensively used for estimating standard Gibbs energy change. Here, we introduce a molecular fingerprint-based thermodynamic tool, dGPredictor, that enables distinguishing between (stereo)isomers in metabolic reactions leading to improved reaction coverage and comparable prediction accuracy as GC methods. dGPredictor can also be used alongside de novo pathway design tools to ensure the correct directionality of

**Funding:** This work is supported by the Center for Bioenergy Innovation (CBI), a U.S. Department of Energy Bioenergy Research Center supported by the Office of Biological and Environmental Research in the DOE Office of Science under contract DE-AC05-00OR22725 awarded to CDM and National Science Foundation funded Molecule Maker Lab Institute (MMLI), award number 2019897 supported by National AI Research Institutes Program of the Directorate for Computer and Information Science and Engineering (CISE), in collaboration with the Division of Chemistry (CHE) and the Division of Chemical, Bioengineering, and Environmental Transport Systems (CBET) awarded to CDM. The funders had no role in study design, data collection and analysis, decision to publish, or preparation of the manuscript.

**Competing interests:** The authors have declared that no competing interests exist.

chosen reaction steps. We applied and tested dGPredictor on reactions from the KEGG database and applied it to screen an isobutanol synthesis pathway design. An open-source, user-friendly web interface is provided to facilitate easy access for standard Gibbs energy change of reactions at different pH values. (https://github.com/maranasgroup/dGPredictor).

## Introduction

Thermodynamic imperatives affect both the direction of metabolic reactions inside the cell and the amount of enzyme needed. There has been a number of efforts aimed at predicting the standard Gibbs energy of metabolites and reactions from compilations of experimental data [1–3]. Standard Gibbs energy estimates can be used in conjunction with metabolite concentration values to infer reaction directionalities[3, 4]. Such analysis tools have been integrated [3] with genome-scale metabolic models to safeguard against the use of reactions in the wrong direction and eliminate thermodynamically infeasible cycles [3, 5, 6].

Gibbs energies quantify thermodynamic constraints and determine both the directionality and efficiency of enzymatic reactions, thereby dictating allowable metabolic phenotypes for product synthesis. For example, the beta-oxidation pathway can be reversed because the overall standard Gibbs energy change in the reverse direction becomes negative when utilizing ferredoxin as the reducing equivalent [7]. This engineered reversed pathway can be used to produce higher-chain linear alcohols and fatty acids with greater energy efficiency [7]. The biological methanogenic and acetogenic reduction pathways are highly efficient in converting $CO_2$ to $CH_4$ due to lower thermodynamic barriers compared to the corresponding geochemical pathways [8]. Thermodynamic analyses are an important tool for assessing and selecting feasible heterologous metabolic pathways [5] and quantifying the thermodynamic driving force for biosynthesis in different production hosts using intracellular metabolomic data [9]. However, direct experimental measurements of standard Gibbs energy change of reactions ($\Delta_r G'^o$) are still limited to ~600 enzymatic reactions cataloged in the Thermodynamics of Enzyme-catalyzed Reactions Database (TECRDB) [5]. Emerging isotopic labeling experiments such as deuterium-labeled studies [10] can directly quantify $\Delta_r G'^o$ of enzymatic reactions but have so far been limited to central carbon metabolism thus necessitating the use of predictive computational frameworks.

In order to expand the prediction of $\Delta_r G'^o$ for reactions lacking experimental measurements, group contribution (GC) methods were developed. They assume that the $\Delta_r G'^o$ can be expressed as the sum of contributions from all functional groups ($\Delta_g G^o$) (based on a predefined list of substructures [2]), which in turn are fitted from experimental data. Multiple linear regression is typically applied to determine each group's contribution in a reaction by minimizing the mean squared error between the observed and estimated $\Delta_r G^o$. Various group contribution methods were developed to improve the prediction accuracy [11]. Table 1 contains a brief description of existing methods for $\Delta_r G'^o$ estimation.

Despite the extensive efforts to improve group contribution predictions, there is a number of inherent limitations [11]. The expert-defined groups provide incomplete coverage leading to (i) metabolites that cannot be decomposed, and thus their $\Delta_f G^o$ cannot be estimated, (ii) an assignment of zero for $\Delta_r G^o$ of reactions with no group changes despite experimental values being non-zero (such as isomerase reactions), and (iii) large uncertainties in $\Delta_r G^o$ for reactions that involve groups occurring sparingly in the training dataset.

**Table 1. Existing methods for the prediction of standard Gibbs energy of biochemical reactions.**

| Algorithm/Method | Citation | Major contributions |
|---|---|---|
| Group contribution | Mavrovouniotis et al. [2] (1990) | • Estimate Gibbs energy of formation by decomposing compounds into functional groups based on biochemical knowledge |
| Group contribution | Jankowski et al. [3] (2008) | • Gibbs energy estimation for biochemical reactions in *Escherichia coli* metabolic network and improved previous method by introducing additional groups and interaction factors |
| Group contribution | Noor et al. [12] (2012) | • Improved last method by integrating known thermodynamic data for molecules when available<br>• Considered pseudoisomers of molecules at different protonation levels to accounts for the effect of pH and ionic strength |
| eQuilibrator | Flamholz et al. [13] (2012) | • User-friendly web-interface based on previous group contribution method by Noor et al. [12] |
| Component contribution | Noor et al. [14] (2013) | • Prioritize on reactant contribution over group contribution, which directly uses the formation energy when available |
| Updated group-contribution | Du et al. [15] (2018) | • Improved previous methods using entropy and enthalpy information<br>• Considered influence of magnesium binding with metabolites and temperature |

Moving beyond simple molecular groups, more than 11,145 descriptors have been developed for extracting chemical information for describing Quantitative Structure-Activity Relationships (QSAR) [16]. Molecular descriptors such as molecular [17] and circular fingerprints [18] use fragments/moieties to represent substructure information as vectors, matrices, or other mathematical representations such as encoding group information in GC methods. They were applied extensively, alongside statistical and machine learning algorithms, to predict physical and (bio-)chemical properties (such as dissociation constant, viscosity, and toxicity) for drug design [19]. However, their application for $\Delta_r G'^o$ estimation of enzymatic reactions has been limited to only a few studies [20, 21]. The computation tool IGERS [20] predicts $\Delta_r G'^o$ of a new reaction based on its similarity (calculated using 2D molecular descriptors) to reference reactions with experimental measurements. More recently, a machine learning algorithm using chemical fingerprints as a feature [21] showed improved prediction accuracy.

Chemical substructures have been used extensively to encode molecular information in many computational *de novo* pathway design tools [22]. Substructure changes in enzymatic reactions can be generalized as rules, thereby codifying *de novo* reactions to fill in missing chemical conversion steps. However, current *de novo* pathway design tools allow only *a posteriori* evaluation of thermodynamic feasibility of a proposed metabolic pathway as novel reaction steps are generally treated as being reversible [23]. Therefore, significant computational resources may be expended in generating pathways with one or more steps operating in a thermodynamically unfavorable direction. Even though tilting the relative reactant/product concentrations can maintain feasibility of the designed pathway, the imposed concentration ranges may not be physiologically viable. Furthermore, operating near thermodynamic equilibrium dramatically increases the required enzyme level, incurring a significant metabolic burden [6, 14]. Thus, an automated approach for estimating the $\Delta_r G'^o$ of all novel steps is required, which can also be easily embedded within pathway design tools such as novoStoic [24]. This would help refine *de novo* predictions by constraining the reaction steps/rules in only the thermodynamically feasible direction.

Herein, we developed a moiety-based automated fragmentation tool called dGPredictor for $\Delta_r G'^o$ estimation of enzymatic reactions. Moieties are descriptors of the bonding environment of all non-hydrogen atoms in a chemical structure. They differ from functional groups which fragment molecules into their constitutive parts [25]. In contrast, moieties are descriptors of all non-hydrogen atoms in a molecule encoding their bonding environment up to a pre-specified bonding distance. Multiple models were tested within dGPredictor using moiety descriptions of different spans and using both explainable linear regression and neural network-based

**Table 2. Details of prediction accuracy and cross-validation scores for different regression models.**

| Model | Mean squared error (kJ/mol)$^2$ | cross-validation: Mean absolute error (kJ/mol) |
|---|---|---|
| M1-linear | 38.30 | 5.83 |
| M2-linear | 24.60 | 15.46 |
| **M1,2-linear** | **9.60** | **5.48** |
| M1-nonlinear | 20.81 | 6.64 |
| M2-nonlinear | 6.92 | 14.69 |
| M1,2-nonlinear | 6.95 | 7.27 |
| **GC** [14] | **45.20** | **5.32** |

nonlinear formalisms. Resultant prediction accuracies and overfitting potential were calculated as the mean squared error (MSE) over training data and median of mean absolute error (MAE) of leave-one-out-cross-validation (LOOCV) results, which were used as two metrics to compare the prediction accuracies for $\Delta_r G'^o$ estimates and facilitate the direct comparison with widely used GC method (specifically, component contribution (CC) [14]). dGPredictor improved the goodness of fit over the widely used GC method [14] by 78.76% (i.e., MSE over training data from the TECRDB database (see Table 2)). It also led to an increase in the coverage of $\Delta_f G'^o$ and $\Delta_r G'^o$ estimation for metabolites and reactions present in the KEGG database by 17.23% and 102%, respectively, over GC by allowing for stereochemical considerations not captured in previous expert-defined chemical groups. Examining the sensitivity of model predictions to moiety definition revealed that moieties spanning a bonding distance of two are more prone to overfitting as compared to the moieties of binding distance one due to the combinatorial explosion of unique moiety types (e.g., cross-validation MAE of 5.83 kJ/mol vs. 15.46 kJ/mol for bonding distance one and two, respectively). Select moieties spanning distances one and two in a combined linear model were found to be less prone to overfitting than the corresponding non-linear variants (cross-validation MAE 5.48 kJ/mol vs. 7.27 kJ/mol for the best performing non-linear NN model). This is likely due to the relatively small size (i.e., 4,001 reactions and 673 metabolites) of the training dataset that does not seem to benefit from using a more extensive set of descriptors embedded in a non-linear modeling framework.

Next, we employed dGPredictor to aid metabolic pathway design by improving upon the current practice of considering *de novo* reactions as being always reversible, thereby necessitating additional scrutiny to ensure thermodynamic feasibility. We show that dGPredictor can estimate the $\Delta_r G'^o$ of *de novo* reactions (i.e., involving novel metabolites) by utilizing their chemical structure information in the IUPAC International Chemical Identifier (InChI) strings [26]. Furthermore, pathway design tools can use the moiety changes in dGPredictor as reaction rules to ensure thermodynamic feasibility.

## Results

### Automated fragmentation of metabolites

In contrast to the widely used GC methods that rely on expert-defined groups, we apply an automated fragmentation approach similar to Carbonell et al. [17] to encode novel molecules. dGPredictor classifies every (non-hydrogen) atom in a structure by assessing its bonding environment at a distance of one (M1 moieties) or two bonds (M2 moieties). We do not consider bonding distance zero (i.e., M0 moieties) as that is equivalent to encoding each atom as a moiety. Bonding distances higher than two were also evaluated but did not yield a significant performance increase, as the available regression dataset was relatively limited in size.

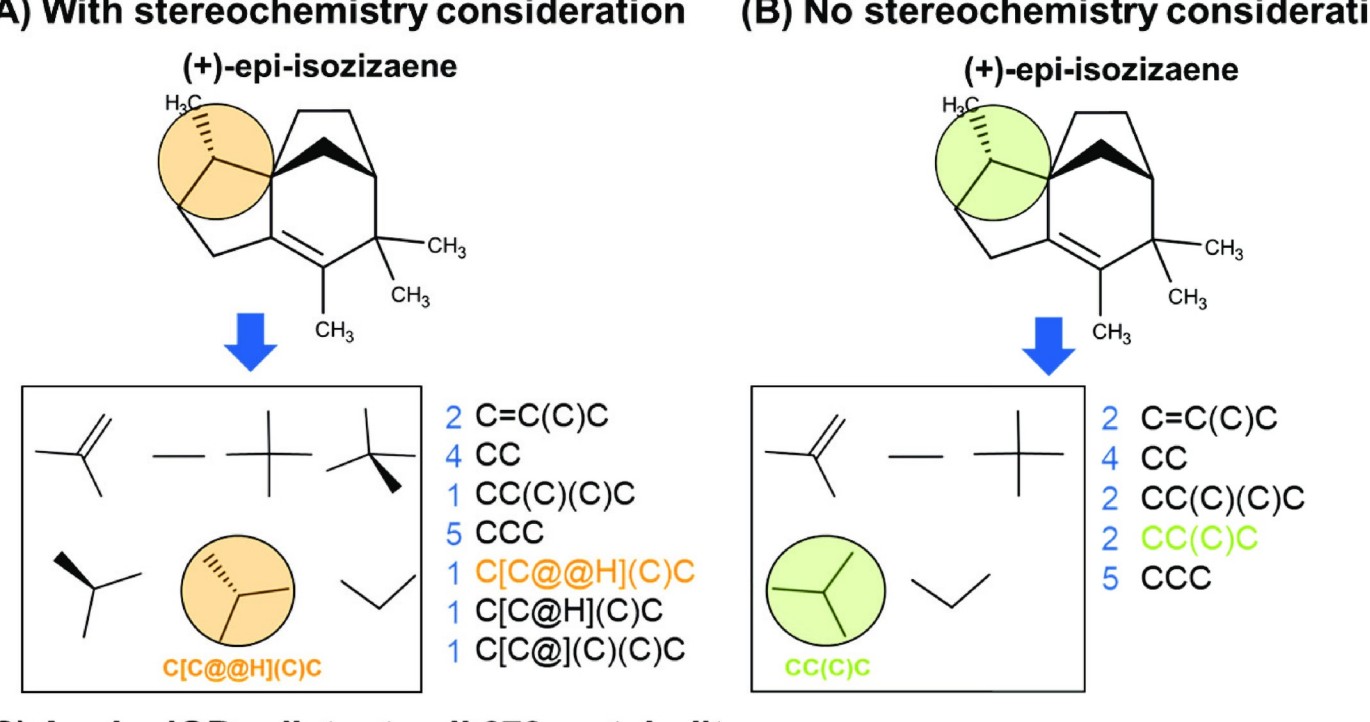

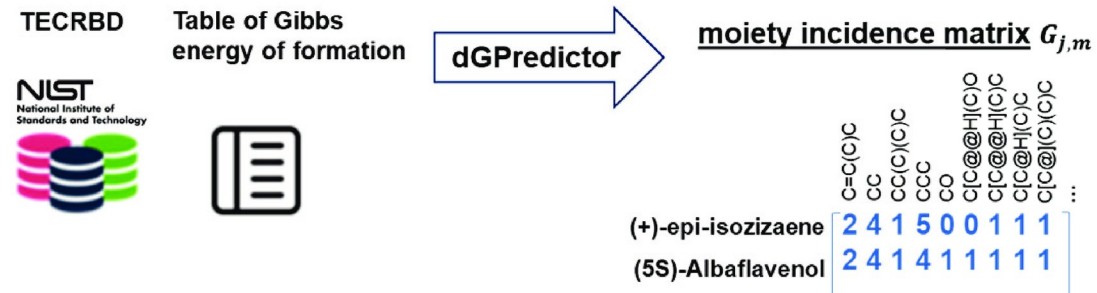

**Fig 1. The substructures/chemical moieties generated by dGPredictor.** The decomposition of (+)-epi-isozizaene (A) with stereochemistry consideration and (B) without stereochemistry consideration, and (C) moiety incidence matrix created by dGPredictor for all metabolites in TECRDB.

Fig 1A and 1B highlight the possible loss of stereochemical information that may occur when not considering the two different configurations around an asymmetric tetrahedral atom, which results in the same moiety description for both configurations. We use (+)-epi-isozizaene as an example to demonstrate the chemical moieties-based metabolite descriptors used in dGPredictor. Fig 1A shows the seven chemical moieties generated from (+)-epi-isozi-zaene when stereochemistry is considered. The occurrence of the seven moieties is counted and used as the moiety incidence matrix $G_{i,g}$ (shown in blue, where rows denote moieties, and the columns count the occurrence of each moiety in the chemical structure and corresponding Simplified molecular-input line-entry system (SMILES) representation). The moiety shown in orange was obtained from the SMILES annotation C[C@@H](C)C, where the chiral specifica-tion (i.e., "C@@H") indicates the carbon atom is the tetrahedral center. The two different sym-bols, "@" and "@@" indicate anticlockwise and clockwise configurations, respectively.

Without stereochemistry, (+)-epi-isozizaene can be decomposed into only five moieties (see Fig 1B). The moiety shown in green is now represented by the SMILES string CC(C)C without chiral specifications. Thus, including stereochemistry-based decomposition within dGPredictor increases resolution by describing moiety changes for reactants and products that differ only in stereochemistry (e.g., isomerases).

The component contribution method [14] cannot decompose (+)-epi-isozizaene due to the incomplete coverage of substructures. As shown in Fig 1, dGPredictor can encode metabolites with either very complex structures or small molecules such as hydroxylamine ($NH_2OH$). We applied dGPredictor to construct the moiety incidence matrix for the 673 metabolites with experimental measurements in TECRDB (see Fig 1C). For moieties spanning a bonding distance of one and two, we generated 263 (S1 Table) and 1,380 (S2 Table) members, respectively, using the RDkit python package (See Methods section 'Automated fragmentation of metabolites').

## Improved goodness of fit of $\Delta_r G'^o$ by dGPredictor using an automated moiety classification method

The Gibbs energy contribution of moieties for predicting overall $\Delta_r G^o$ of reaction was generated separately using Bayesian ridge regression [27, 28] and feed-forward neural networks [29, 30]. Moiety generation and the regression models employed are described in detail in the Method section. Note that as experimental data was collected at different pH and ionic strengths, a mixture of pseudoisomers (i.e., multiple structures with different protonation states) exists for each metabolite. We used the Inverse Legendre Transform (see details in Methods section 'Pseudoisomer') to standardize the experimental values of $\Delta_r G^o$ to a single pseudoisomer [31]. The Inverse Legendre Transform reduces the ensemble of pseudoisomers to a single pseudoisomer (i.e., the most abundant pseudoisomer at pH 7 and ionic strength 0.25M) which was used as the reference for regression analysis in this section. The difference between $\Delta_r G^o$ calculated for a single pseudoisomer and the transformed Gibbs energy $\Delta_r G'^o$ (i.e., Gibbs energy of the ensemble) at a specific cellular condition is corrected as a function of the dissociation constant $pKa$ for each pseudoisomer, pH, and the ionic strength when making predictions (see details in Methods section 'Pseudoisomers').

First, we applied Bayesian ridge regression to estimate the individual contribution of each moiety using experimental measurements of $\Delta_r G^o$ from 4,001 reactions. We considered moieties associated with a bonding distance of one and two to select the model with better prediction accuracy. In the remainder of the paper, we refer to these models of distance one and two as M1-linear (263 unique moieties) and M2-linear (1,380 moieties). In addition, we examined the efficacy of using moieties of both distances simultaneously within a single model (M1,2-linear model, with 1,643 moieties). The mean squared error (MSE) over the entire training dataset was used as a metric for the goodness of fit in models. For the M2-linear model, MSE improved by 35.77% over the M1-linear (see Table 2), which is expected as the number of moieties increases >5-fold from 263 to 1,380. We found that combining moieties of radius one and two in model M1,2-linear further lowered the MSE by 60.97% compared to the M2-linear. Thus, the overall model ranking using the MSE is M1,2 > M2-linear > M1-linear.

Next, we determined the overfitting potential of the constructed models by evaluating the leave-one-out cross-validation mean absolute error (MAE). A large MAE value on the validation data alongside a low MSE value on the training data indicates model overfitting. MAE is the absolute difference between the experimentally observed Gibbs energy change of a reaction (i.e., $\Delta_r G^o_{obs}$ in the validation dataset) and the predicted value from the linear regression model trained without the experimental data of that particular reaction (i.e., $\Delta_r G^o_{est}$ in the training

dataset). We found a significant increase in MAE for M2-linear compared to M1-linear and M1,2-linear models, with a median MAE of 15.46 kJ/mol, 5.83 kJ/mol, and 5.48 kJ/mol, respectively (see Table 2). The M1,2-linear model had the lowest MSE on the training dataset with a value of 9.60 $(kJ/mol)^2$ compared to 38.30 $(kJ/mol)^2$ for M1-linear and 24.60 $(kJ/mol)^2$ for M2-linear. Therefore, we chose the M1,2-linear as the best among the three linear models due to the lowest overfitting potential and highest cross-validation prediction accuracy. Do note that this should not be interpreted as a result that would hold universally for all datasets, and this analysis should be performed before selecting an appropriate moiety bonding distance.

## Introducing neural network-based nonlinear moiety contribution leads to a marginal increase in prediction performance

We next explored whether the inherent linearity of moiety contributions in the developed models limits their predictive capability. We used a feed-forward multi-layer neural network to allow for a nonlinear description of moiety contribution for predicting $\Delta_r G^o$. Similar to linear models, three different models with moiety spanning distances one and two were generated to determine the effect of moiety description on prediction accuracy - M1-nonlinear, M2-nonlinear, and M1,2-nonlinear. The non-linear models followed a similar trend as their linear counterparts in terms of goodness of fit scores, i.e., MSE decreases as we increase the number of moieties with distance two due to overfitting (see Table 2). Models M2-nonlinear and M1,2-nonlinear were nearly identical in performance on training dataset (MSE 6.92 $(kJ/mol)^2$ vs. 6.95 $(kJ/mol)^2$) and outperformed the M1-linear model (>3-fold higher MSE 20.81 $(kJ/mol)^2$). However, the M1,2-nonlinear model performs well in terms of both MSE and LOOCV MAE scores, indicating that the M1,2-nonlinear is the better-performing nonlinear model.

A comparison among the six models (three linear and three nonlinear) indicates that the MSE values (i.e., prediction accuracy) for all three nonlinear models were better than their linear counterparts. However, a correspondingly higher LOOCV MAE reveals that the MSE improvement is likely due to overfitting. For example, M1,2-nonlinear has a slightly lower MSE compared to the M1,2-linear (6.95 $(kJ/mol)^2$ vs. 9.60 $(kJ/mol)^2$). However, cross-validation scores showed that the M1,2-linear is significantly less susceptible to overfitting with MAE 5.48 kJ/mol vs. 7.27 kJ/mol. This implies that the extra complexity and lack of interpretability associated with the M1,2-nonlinear model do not come with any appreciable increase in prediction performance. Since the MSE of both models is comparable, we chose model M1,2-linear as the dGPredictor default model in all subsequent results that were computed using this model. The M1,2-linear model also offers straightforward interpretability of the obtained Gibbs energy predictions.

Finally, we assessed model performance between the automated decomposition scheme using moieties proposed herein and expert-based groups. We applied the same set of thermodynamics data compiled by Noor et al. [14] (available at https://github.com/eladnoor/component-contribution/) on dGPredictor and the GC method to enable direct comparison. dGPredictor using the M1,2-linear model outperformed GC, with a lower MSE (9.60 $(kJ/mol)^2$ vs. 45.20 $(kJ/mol)^2$) and again near perfect $R^2$ score (0.9998 vs. 0.9989) (Fig 2A) estimated over the entire training dataset. As evaluated by Du et al. [11], the linear regression-based GC model often predicts unrealistically large $|\Delta_g G^o|$ for functional groups with limited representation in the training data. A similar observation was also made in the current study wherein GC predicted a $|\Delta_g G^o|$ value greater than 1,500 kJ/mol for two groups (Fig 2B), while the largest value in dGPredictor is ~200 kJ/mol. dGPredictor mitigates this issue by applying

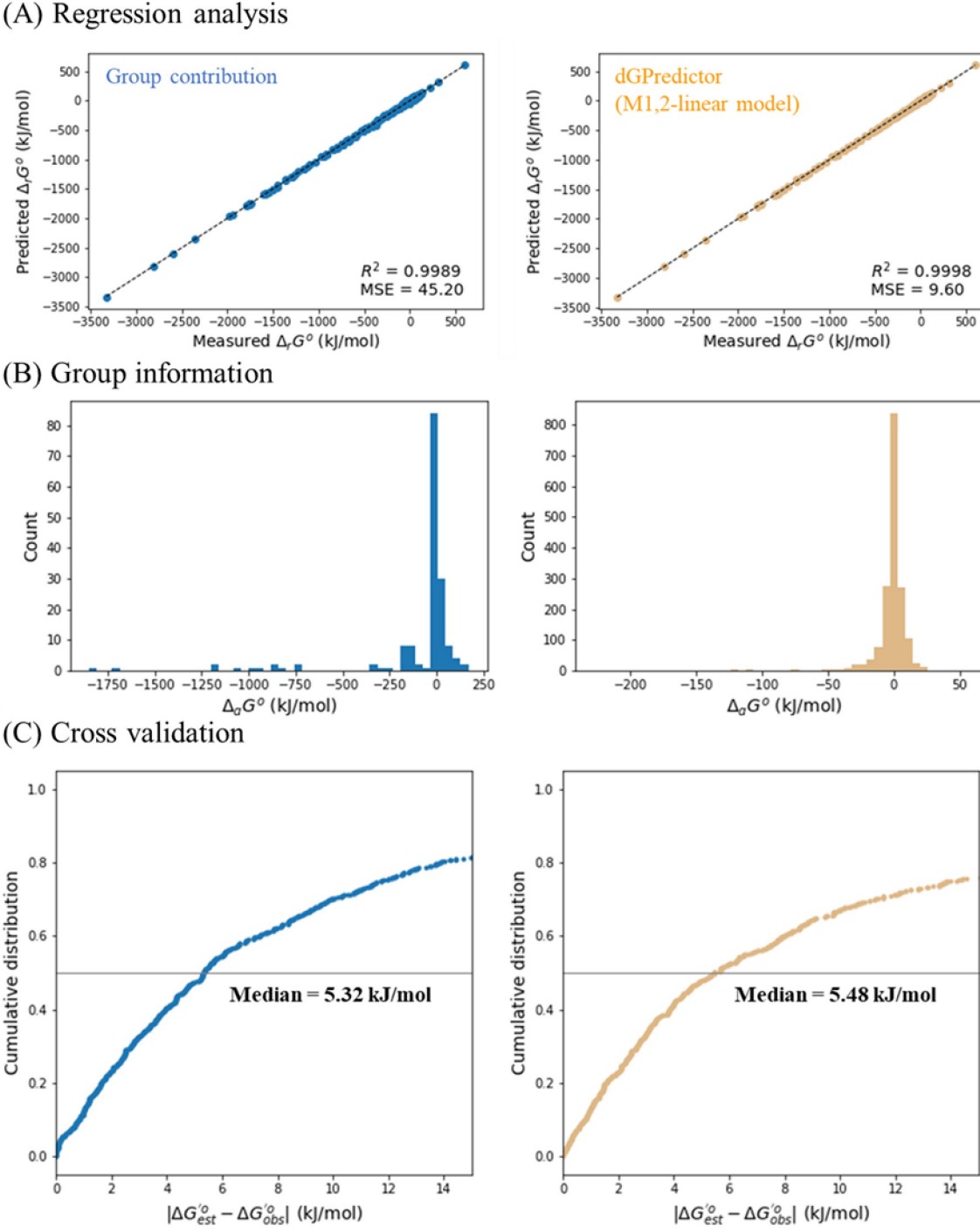

**Fig 2. Comparison between group contribution method by Noor et al. [14] and dGPredictor (best model: M1,2-linear).** (A) Comparison of regression analysis for the best dGPredictor model (M1,2-linear model) and previous group contribution method. It shows the improvement in the mean squared error (MSE improved from 45.20 $(kJ/mol)^2$ to 9.60 $(kJ/mol)^2$ and nearly identical R-squared values. (B) Histogram denoting the estimated contribution of groups. dGpredictor group contribution estimates do not predict unrealistically high values, unlike group contribution. (C) leave-one-out cross-validation results, which indicates very close performance with the previous method with an improved MSE value.

regularization in the regression analysis, which lowers moiety contribution estimates by adding L2-regularization term in the regression objective function which also helps avoid overfitting (see Methods section for details). It must be noted that the median MAE from GC and dGPredictor is 5.32 kJ/mol and 5.48 kJ/mol respectively (Fig 2C) indicating similar overfitting potentials.

## Increased coverage of metabolites and reactions by dGpredictor

A rigorous test of any predictive model lies in its ability to generalize over unseen data. Thus, we used the KEGG database [32] to test the dGPredictor's scope and prediction capability for reactions that are not included in the training dataset. This also helps evaluate the model's ability to provide genome-scale coverage and estimate $\Delta_r G^o$ for novel reactions when designing *de novo* biochemical pathways. The KEGG database instance used consisted of 15,278 metabolites and 7,053 reactions with chemical structure information defined by InChI strings. We first compared the ability of the GC method and the automated fragmentation method of dGPredictor to describe all metabolites present in the database as chemical functional groups and moieties. The GC method could describe 85.3% of the 15,278 metabolites, while dGPredictor succeeded in describing all metabolites (see Table 3). The GC method missed 14.7% metabolites because they contain unique substructures that are not included in its expert-defined 163 groups. Fig 3A illustrates a few of these unique moieties and the corresponding dGPredictor decomposition. Most of the substructures that were not covered in GC methods decomposition involved bonds with N, P, and S atoms. Next, the $\Delta_r G^o$ of reactions in the KEGG database was calculated using dGPredictor and GC. Metabolites from the TECRDB database [33] were decomposed using expert-defined GC groups and the automated moiety-based framework in dGPredictor and used to calculate $\Delta_r G^o$. GC could successfully describe 33.8% of the database reactions, while dGPredictor could parameterize twice that number (i.e., 69.3%), indicating that the developed formalism can successfully generalize to provide larger reaction coverage.

Even though dGPredictor improved the coverage of KEGG metabolites and their corresponding reactions, there are still metabolites with associated moieties absent from the TECRDB dataset, leading to incomplete moiety coverage during model training. For example, the reaction shown in Fig 3B represents six moiety changes. However, the moiety "C-N-N" is not in the list of the 1,643 moieties generated from metabolites in the TECRDB training dataset (for both radius one and two). Noor et al. [14] flagged all these instances by setting a standard deviation of $10^{10}$kJ/mol to indicate that it cannot estimate $\Delta_r G^o$ with any level of certainty. dGPredictor uses Bayesian ridge regression to provide (generally) a narrower $\Delta_r G^o$ range by assuming an isotropic Gaussian distribution with precision parameter $\alpha$ and identity matrix $I$ for the standard Gibbs energy contributions of the moieties/groups ($\Delta_g G^o$):

$$p(\Delta_g G^o | \alpha) = N(0, \alpha^{-1} I)$$

After the hyperparameter optimization during model training (see Methods section 'Bayesian ridge regression to determine the Gibbs energy of groups $\Delta_g G^o$') to estimate parameter $\alpha$

**Table 3. Statistics of 15,278 metabolites and 7,053 reactions from KEGG that can be decomposed into groups/moieties.**

|  | GC | dGPredictor |
|---|---|---|
| Coverage of KEGG metabolites | 85.3% (13,032/15,278) | 100% (15,278/15,278) |
| Coverage of KEGG reactions | 33.8% (2,385/7,053) | 69.3% (4,887/7,053) (using moieties from TECRDB training data) <br> 100% (using moieties from KEGG metabolites) |

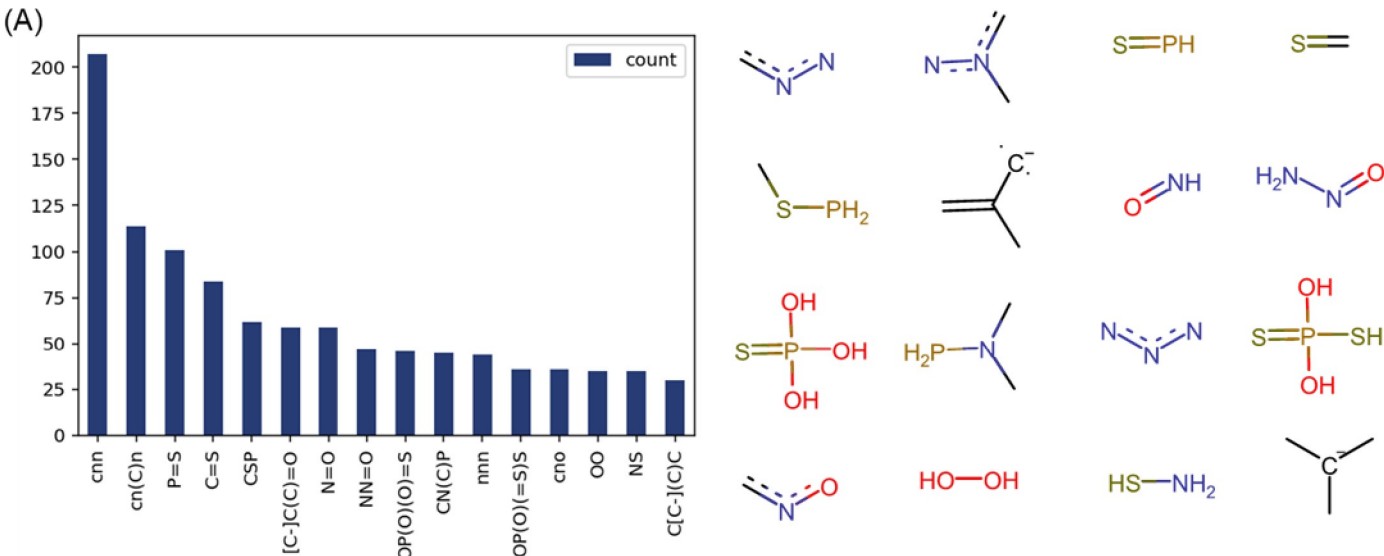

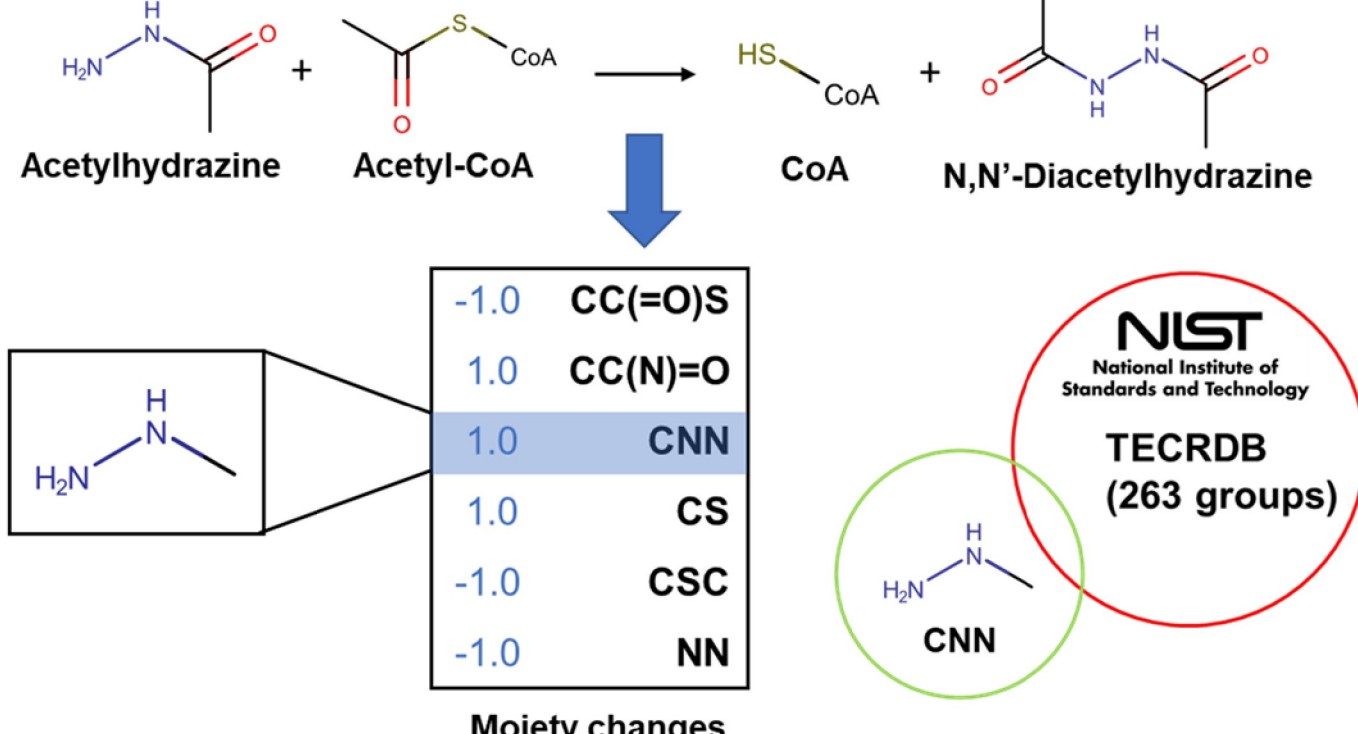

**Fig 3. Unique substructure covered by dGPredictor.** (A) Unique substructures in the 14.7% metabolites missed by previous GC method. Here the count indicates the occurrence of such substructures in metabolites (B) Example of a reaction with groups (i.e., group "C-N-N") missing from the 263 groups from TECRDB.

[28] by maximizing the log of the posterior distribution in Bayesian ridge regression yields a parameter $\alpha$ of 9.023 $10^{-4}$. The standard deviation of $\Delta_g G^o$ in dGPredictor is computed as $\sigma = \sqrt{\alpha^{-1}} = 33.29$ kJ/mol. As shown in Fig 2B, the $\Delta_g G^o$ contributions of 1,643 moieties are well-determined as their estimates for 99% fall within three standard deviations of their mean.

Therefore, Bayesian ridge regression provides a much narrower confidence interval for all moieties with no prior thermodynamic information. The model predictions can thus increase to 100% reaction coverage with Bayesian regression, although with a wider confidence interval in $\Delta_r G'^o$ for reactions that contain moieties without experimental data. Hence, the use of an automated moiety-based description with Bayesian regression allows expanding the set of Gibbs energy predictions while narrowing the overall confidence interval as compared to the previous GC method.

## dGPredictor enhances prediction scope by accounting for reactions with no GC-defined group changes

A limitation of previous GC methods lies in their inability to predict $\Delta_r G'^o$ of reactions with no group changes, which are ultimately assigned a $\Delta_r G'^o$ value of zero. Many of these reactions are isomerases or transaminases, which may have non-zero $\Delta_r G'^o$ [11]. One such example is the enzymatic reaction catalyzed by GDP-D-mannose 3,5-epimerase (EC number: 5.1.3.18), with an experimentally measured equilibrium constant $K' = 1.94$, implying a $\Delta_r G'^o$ of 1.7 kJ/mol [34]. The reactant and product, GDP-mannose, and GDP-L-galactose, respectively, are structural isomers (Fig 4A), due to which GC-based methods are unable to capture a group change in the reaction. dGPredictor accounts for the inherent stereochemical changes by capturing the clockwise and anticlockwise configurations of the two chiral centers (Fig 4A). The predicted $\Delta_r G'^o$ is 1.74 kJ/mol, which is quite close to the experimental measurement of 1.7 kJ/mol.

The KEGG database has 319 reactions associated with no group changes, as defined by GC-based methods (Fig 4B). Most of them are transferases (EC 2) and isomerases (EC 5), which cannot be captured due to no group changes from the 163 expert-defined groups because of its inability to differentiate stereochemistry in chemical structures. dGPredictor described 86.83% of the reactions with no group changes (i.e., 277/319) and, in particular, 39.71% isomerases (i.e., 110/277). This is because the simultaneous inclusion of moieties spanning distances one and two allows us to consider additional details of the localized bonds and atoms, thus alleviating problematic cases of no moiety changes being registered when considering single bonding distance. For example, the moiety description of an aminotransferase reaction (EC 2) (Fig 4C) generates identical moieties for both substrates and products, leading to an empty group change vector when considering bonding distance one moieties. However, allowing an additional bonding distance of two helps dGPredictor generate unique moieties, thus resolving the issue of zero change and in turn leading to a non-zero $\Delta_r G'^o$ value. We estimated the $\Delta_r G'^o$ for only reactions with no-group changes and found an MSE of 5.06 (kJ/mol)$^2$ for dGPredictor compared to 63.97 (kJ/mol)$^2$ for GC methods (see S3 Table for no-group change reactions in training dataset). The cross-validation error followed the same trend as the mean MAE error for the GC method was 3.84 kJ/mol compared to 2.26 kJ/mol for the dGPredictor (see S3 Table for no-group change reactions in validation dataset). S4 Table is provided with the list of all no-group change reactions from the training dataset resolved by dGPredictor, along with their corresponding experimental and estimated Gibbs energies.

Nevertheless, 42 reactions could still not be resolved because both M1 and M2 descriptions used by dGPredictor were identical between reactants and products. This is because the RDkit tool [35] used for computing moieties stores molecules with implicit hydrogen (i.e., not explicitly present in the molecular graph) and ignores the "C-H/O-H" bond when generating the SMILES string. Thus, RDkit cannot identify an atom to be chiral unless it has all the atoms different at the specific bonding distance that is being considered. We have used moieties of up to distance 2 (i.e., M2). Moieties associated with longer bonding distances tend to significantly

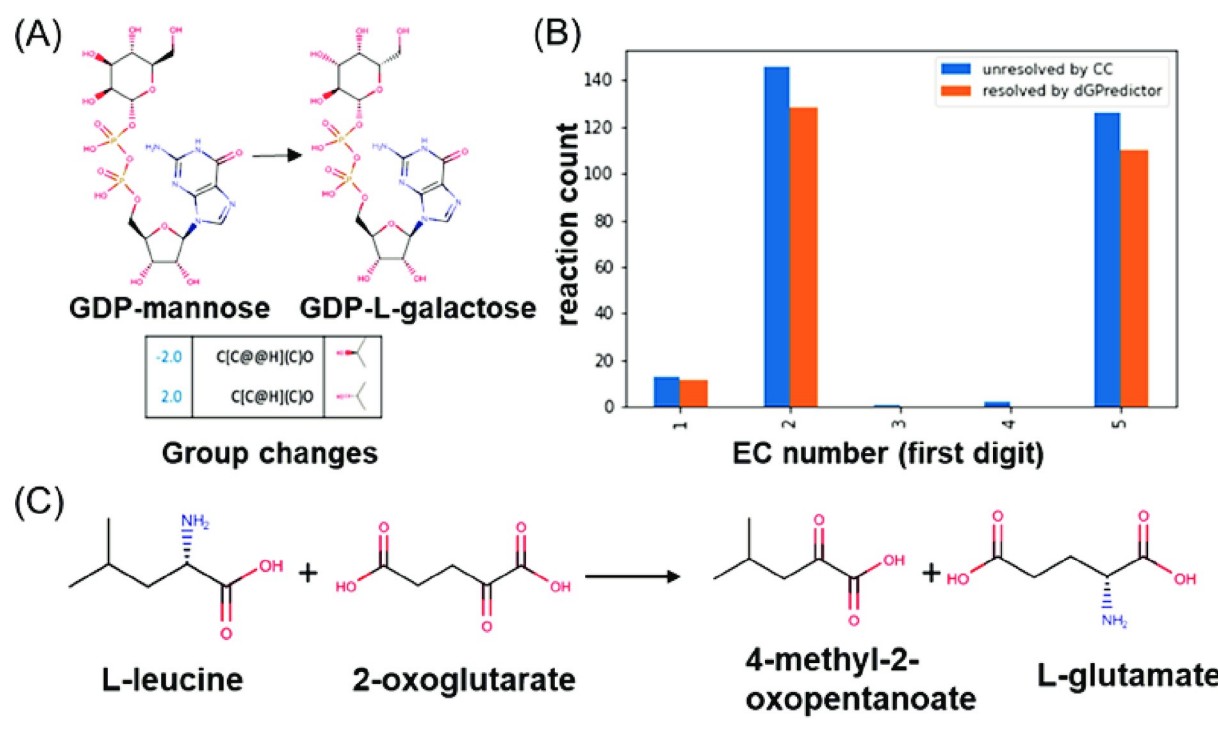

**Fig 4. Increased coverage of reactions with no GC-defined group changes.** (A) GDP-mannose 3,5-epimerase (R00889) reaction that can be resolved to give group change using stereochemistry information, and examples of reactions in EC 2, (B) The number of reactions that have no group changes defined by GC (blue) and resolved by dGPredictor (orange), (C) an example of reaction in EC 5 (D) that cannot be resolved by dGPredictor using only bonding distance one.

increase in number and cause overfitting. An example is shown in Fig 4D, where KEGG reaction R10764 that converts *α*-L-Fucopyranose to *β*-L-Fructopyranose is expected to have non-empty moiety changes if the moieties preserve the same stereo configurations (i.e., [C@H] or [C@@H]) as *α*-L-Fucopyranose and *β*-L-Fructopyranose, but RDkit cannot determine the moiety as being chiral as the "C" atom is attached to two "O" atom for bonding distance one. In this case, RDkit returning an empty group can be alleviated by combining both distances to capture a non-zero moiety change. This flexible moiety consideration is the primary reason behind dGPredictor providing ~87% more coverage for reactions with no group/moiety change than previous GC methods. The remaining 13% could be tackled using more advanced molecular descriptors such as Neural Graph Fingerprints [18] that account for higher-order interactions instead of local atom/bond information.

## User-friendly interface for Gibbs energy calculation of novel reactions and metabolites beyond KEGG

Advances in computational pathway design have expanded the range of microbial product synthesis to non-natural synthetic molecules and drug precursors, leveraging broad-substrate range enzymes [36, 37] promiscuous activity [38], and even *de novo* enzyme design [39, 40]. However, tools such as novoStoic [4] generally treat novel transformations as reversible, necessitating additional scrutiny to ensure the thermodynamic feasibility of the designed pathway. The ability of dGPredictor to characterize unseen metabolites, and by extension, chemical transformations, accompanied by an automated moiety-based framework enables its use in pathway design tools to eliminate thermodynamically infeasible predictions. In this section, we demonstrate how dGPredictor can predict $\Delta_r G'^o$ for novel reactions which do not span known biochemical reactions using chemical moieties, and a GUI that can help users query the same.

The dGPredictor input format uses KEGG IDs to indicate the substrates and products of a reaction. For example, dGPredictor can recognize 'C00096 < = > C02280' as the reaction "GDP-mannose < = > GDP-L-galactose" (discussed in above section). However, KEGG is only one of many databases consolidating biochemical reactions, has a lower metabolite content [41], and does not capture novel molecules that often show up in the reactions of retrosynthetic metabolic pathways [42]. Therefore, we designed dGPredictor to allow for user-defined chemical structures as an additional input to estimate $\Delta_r G'^o$ for any novel reaction. A user-friendly interface has been developed (https://github.com/maranasgroup/dGPredictor) to facilitate thermodynamic analysis for reactions. Metabolites with known chemical structures can be entered using KEGG IDs and InChI strings are required for molecules not present in the KEGG database. For example, in the *de novo* pathway found by novoStoic for pinosylvin (C01745) degradation, a deoxygenase enzyme catalyzes the first step and produces an aromatic product (see Fig 5). KEGG IDs identify all metabolites except the reduced product 'N00001' in the reaction "C01745 + C00004 < = > N00001 + C00003 + C00001". Here, we use ID "N00001" to represent the novel metabolite 3-Phenethyl-phenol, where N refers to novel. The user can indicate that the reaction involves a novel metabolite (i.e., absent in the KEGG database) by clicking a checkbox and filling in the InChI string placeholder under the stoichiometry of the reaction to describe the atom composition and bonds in metabolite N00001 (see

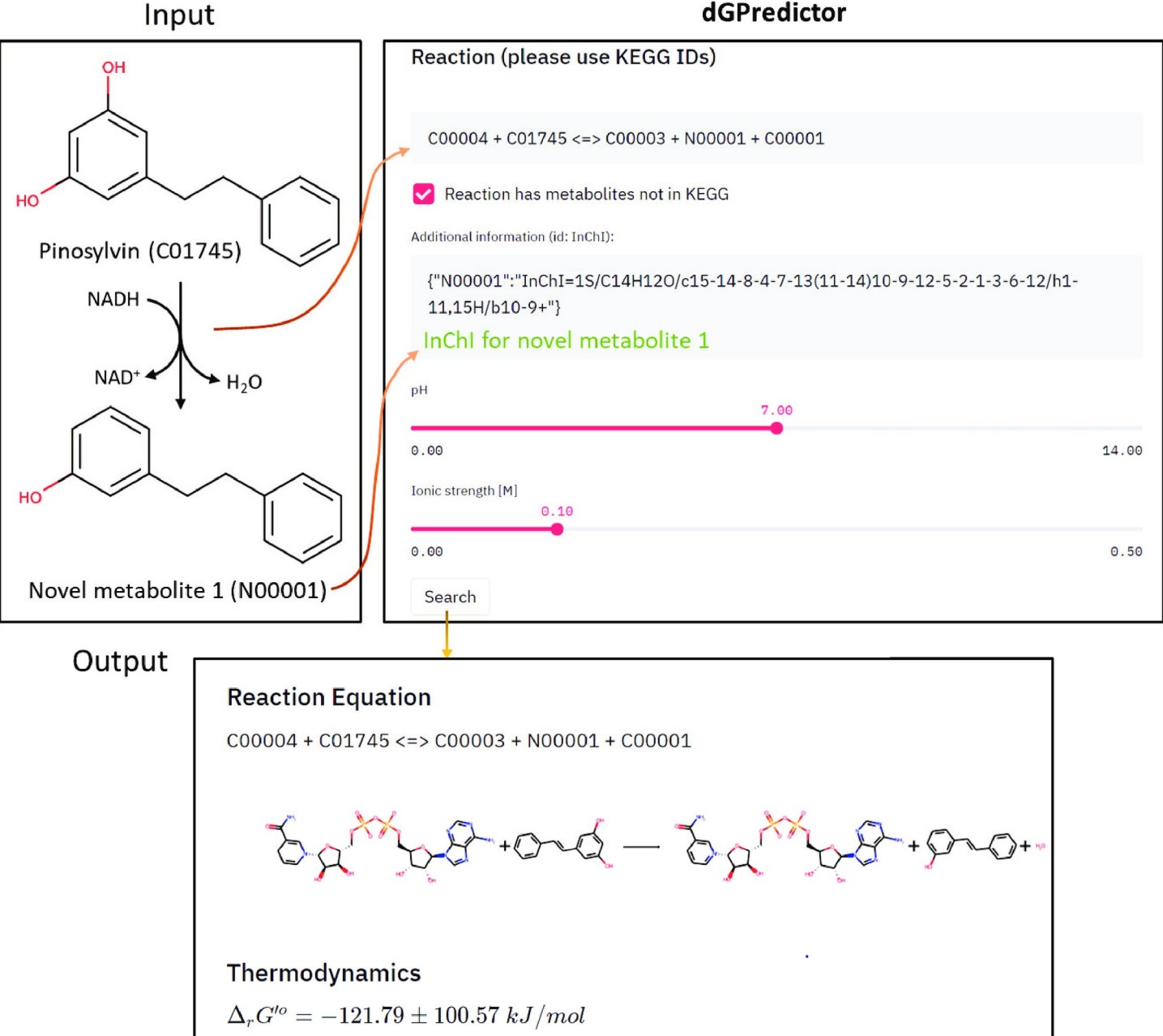

**Fig 5. The graphical user interface of dGPredictor.** It allows the input of metabolites using their KEGG IDs (for known compounds) and InChI strings (for novel compounds) in a chemical reaction. Intracellular conditions of pH and ionic strength can also be adjusted using sliders. The final output shows the standard transformed Gibbs energy $\Delta_r G'^o$ of the reaction at a particular pH and ionic strength.

Fig 5). Similar to eQuilibrator [13], we allow the customized input of intracellular conditions (i.e., pH and ionic strength) via two sliders. With all the information properly defined, on clicking the "search" button, dGPredictor first show the chemical structures in the reaction and then calculate the standard transformed Gibbs energy $\Delta_r G'^o$ of the reaction at a particular pH and ionic strength. The output information also displays the standard deviation of predictions from Bayesian ridge regression and the moiety changes as defined by dGPredictor. Note that the standard transformed Gibbs energy $\Delta_r G'^o$ is not a function of the actual metabolite

concentrations. The measured concentrations of metabolites ($c_i$) can be used to calculate the actual Gibbs energy for a reaction $j$ ($\Delta_r G_j'$) via the equation $\Delta_r G_j' = \Delta_r G_j'^{\circ} + ln(\prod_i [c_i]^{s_{ji}})$ where, $S_{ji}$ is the stoichiometry of metabolite $i$ in reaction $j$. We envision the developed GUI to facilitate easy adoption of dGPredictor to the broader metabolic engineering and synthetic biology community.

## Assigning thermodynamics-derived directionality to reaction rules in *de novo* pathway design

As dGPredictor generates automated molecular descriptions, it can be used in conjunction with pathway design tools to ensure that individual reaction, and in turn, the overall pathway, is thermodynamically feasible. Pathway design tools such as novoStoic, RetroPath, and Retro-Path2.0 [4, 43, 44] can be integrated with the moiety change vector of dGPredictor as reaction rules to design *de novo* pathways. We used the novoStoic tool to illustrate this integration and deployed dGPredictor on the 3,603 unique reaction rules generated from KEGG database. We found that based on the predicted standard free energy of change, a number of novel reactions can reliably be flagged as irreversible (i.e., $|\Delta_r G'^o| > 20$ kJ/mol). We chose a value of 20 kJ/mol because the actual concentrations are rarely known during the pathway discovery stage. Therefore, we used the conservative cutoff of $|\Delta_r G'^o| > 20$ kJ/mol as a threshold for deciding the reversibility of reactions, as reversing reactions with $|\Delta_r G'^o| > 20$ kJ/mol would require highly imbalanced concentrations that are unlikely to be physiologically relevant. This enables the elimination of intermediate steps with a high Gibbs energy barrier, thereby significantly reducing the number of candidate pathways to be explored.

Similar to the definition of reaction rules in novoStoic [4], which captures changes in the topology of molecular graphs for a substrate to product conversion [45], dGPredictor assumes that reactions with same rule undergo same substrate to product change, thereby conforming to identical moiety change. We estimated $\Delta_r G'^o$ for all 3,603 reaction rules and use ±3 standard deviations to obtain confidence intervals (i.e., 99.7% probability that the Gibbs energy estimate is within the calculated interval) (Fig 6A and 6B). We identified 331 reaction rules with predicted Gibbs energy confidence intervals that span only positive values (see Fig 6B) implying that they can only be deployed in the reverse direction. As an additional safeguard, we only applied the irreversibility restriction if the absolute value of the predicted free energy of change $|\Delta_r G'^o|$ exceeds 20 kJ/mol, to account for varying metabolite concentrations that may ultimately tilt reaction directionality [46]. This additional constraint reduces the number of reaction rules that can confidently be treated as irreversible from 331 to 325 (see Fig 6C for EC classification of irreversible reaction rules). Therefore, the dGPredictor framework can help identify the direction *a priori* for irreversible reaction rules and eliminate the thermodynamically infeasible intermediate steps. This allows novoStoic to consider not only the overall thermodynamic feasibility of the pathway but also evaluate every single step.

To assess the extent of reaction rules' directionality affecting pathway design predictions, we applied the set of reaction rules to search for isobutanol production pathways (a well-studied product from the Ehrlich pathway [24, 47–49]) using 2-ketoisovalerate as the precursor. Two distinct production pathways were identified by novoStoic (Fig 6D). Pathway A is the engineered pathway in *E. coli* and *C. thermocellum* with a demonstrated thermodynamic feasibility *in vivo*. Pathway B (shown in orange), however, is thermodynamically infeasible. The first step converts 2-ketoisovalerate to isobutyric acid using a novel reaction R1, which has the same moiety change as the native reaction of indole-3-pyruvate monooxygenase. However, the second step is a novel reaction R2 similar to the reaction catalyzed by nucleoside oxidase that favors the reverse direction based on dGPredictor thermodynamics analysis. Thus, instead of relying on

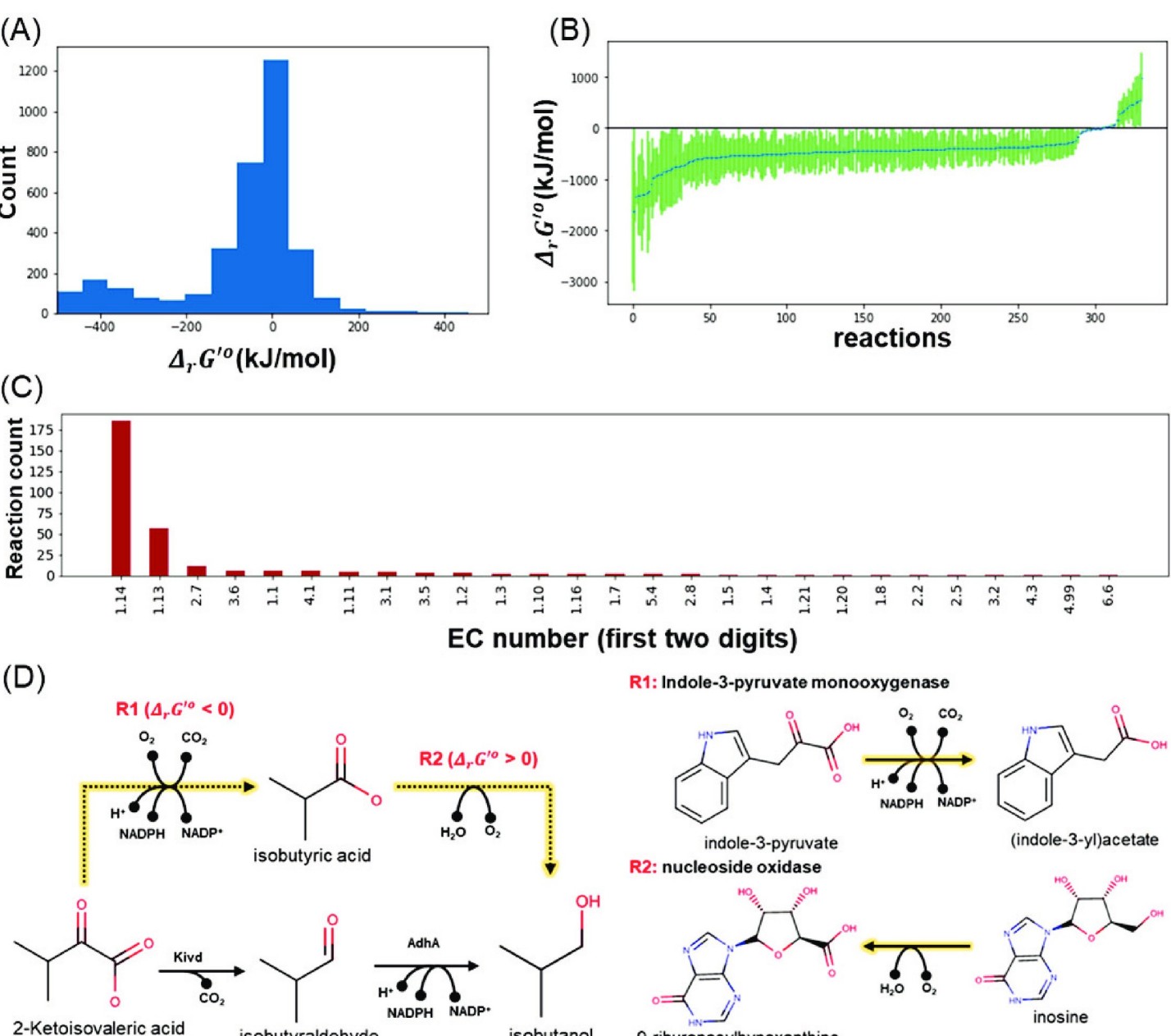

**Fig 6. Moiety/Group change vectors can be used as reaction rules to design *de novo* pathways.** (A) $\Delta_r G'^o$ for all the 3,603 reaction rules, (B) 331 reaction rules found to be irreversible from the uncertainty analysis using ± 3 standard deviations (C) The EC classification for irreversible reaction rules, and (D) an example of thermodynamically infeasible pathways eliminated by using irreversible reaction rules. (See S5 Table for the Gibbs energy information of reaction rules and their corresponding KEGG IDs).

manual post-processing, direct inclusion of thermodynamic constraints from dGPredictor within the novoStoic pathway design tool can help eliminate infeasible pathways.

## Discussion

This paper presents an automated thermodynamic analysis tool, dGPredictor, based on molecular fingerprints and chemical moieties to estimate the $\Delta_r G'^o$ of biochemical reactions.

Compared with previous group contribution methods, dGPredictor expands thermodynamic calculation coverage to more molecules and reactions with improved accuracy. This is primarily due to the automated fragmentation method that allows incorporating stereochemical information while generating chemical moieties, which was lacking in the group decomposition scheme in previous methods [2, 12, 14, 50]. The versatility of the fragmentation method is further illustrated by extending predictions to *de novo* reactions, which enables determining the thermodynamical feasibility of synthetic pathways and thus aid strain design algorithms.

First, we showed that the moiety-based automated fragmentation using the SMILES notations could account for stereochemistry in metabolite descriptions. Moieties spanning bonding distances of one and two were generated and used in a linear and non-linear regression framework to ascertain the best-performing model. We found that employing a non-linear framework and moieties spanning a bonding distance of two leads to a slight increase in prediction accuracy while being significantly more prone to overfitting. Therefore, an explainable linear model comprising moieties spanning bonding distances of one and two (i.e., M1,2-linear) was determined to be the best performing model and the default for dGPredictor. Notably, the cross-validation MAE (representing prediction accuracy and overfitting potential) was comparable to the current state-of-the-art GC method (Component Contribution) and the MSE (metric for goodness of fit) improved by 78.76% over same training (i.e., the TECRDB database) and cross-validation (i.e., the KEGG database) datasets for a direct comparison of model predictions. We found that the proposed automated fragmentation approach can be implemented on all metabolites in the TECRDB database, unlike the GC method, which uses expert-defined functional groups to decompose metabolites and only cover 85.3% of metabolites. Molecular descriptions thus obtained were used to estimate $\Delta_r G'^o$ for reactions in the KEGG database, leading to increased reaction coverage (69.3% vs. 33.8%). However, there remains scope for improvement in the dGPredictor prediction pipeline, which can be aided by increasing the experimental coverage of reactions with experimental $\Delta_r G'^o$ estimates, in turn, increasing the number of characterized moieties. dGPredictor can be used to prioritize these experimental targets for $\Delta_r G'^o$ estimation by focusing on reactions that comprise unique moieties frequently occur in the uncovered reactions. A list of such reactions ranked by the frequency of unmeasured moieties is provided in the supplementary file S6 Table. In addition, quantum chemical calculation [51, 52] that *de novo* estimate $\Delta_f G'^o$ for metabolites with no experimental measurements and/or novel metabolites generated by retrosynthetic pathway design algorithms can be used to supplement the available experimental data.

A graphical user interface was built for the dGPredictor tool, similar to the group contribution-based web interface eQuilibrator. It can estimate $\Delta_r G'^o$ at different intracellular conditions, namely pH and ionic strength. dGPredictor improves upon eQuilibrator by not only estimating the $\Delta_r G'^o$ for reactions with compounds in the KEGG database and allowing the user to input novel metabolites using their InChI strings. Thus, our tool broadens the capability of $\Delta_r G'^o$ prediction by incorporating novel reactions. This prediction ability is extended to estimate the $\Delta_r G'^o$ of designed pathways by restricting the directionality to new reaction rules in the *de novo* pathway design tool novoStoic [4]. dGPredictor can help eliminate unnecessary solutions in novoStoic that are not thermodynamically feasible. The moiety change vectors in the dGPredictor can be directly used as allowed reaction rules in novoStoic [4]. dGPredictor thus facilitates an effective search for thermodynamically feasible metabolic pathways by discarding on average 10% of the reaction rules as infeasible, thereby reducing the search space and computational expense for pathway design.

However, it must be noted that there still exists scope for improvement in the molecular descriptors proposed herein. dGPredictor cannot fully resolve reactions with no moiety changes in isomerase reactions due to limitations in the fingerprints defined by RDkit.

Therefore, applying advanced 3D molecular descriptors to capture the exact stereochemistry might help capture such reactions. 2D molecular descriptors only contain information for the localized atoms/bonds, whereas a 3D descriptor allows precise capture of molecular shape and interactions [16]. In a recent study, the Quantitative Structure-Activity Relationship (QSAR) method based on Smooth Overlap of atomic position (SOAP) descriptors was shown to successfully capture the 3D atomic environment from conformers [53]. Therefore, utilizing a 3D molecular descriptor for molecule decomposition in the free energy prediction has great potential to account for even more accurate estimation. In addition to that, dGPredictor cannot handle reactions involving metals such as Fe or polymeric structures (e.g., glycogen, starch) due to the lack of available thermodynamic information in the training dataset. This can be remedied with the inclusion of additional relevant information in the dataset.

## Methods

### Automated fragmentation of metabolites

The algorithm first classifies every atom in the chemical structures of a metabolite into groups of adjacent atoms within a specified bonding distance (i.e., moieties). The bonding distance defines the bond's proximity to consider from an atom to be described as a chemical moiety. We use the InChI string of metabolites as input. Next, we represent each fragment/moiety with canonical SMILES string. Finally, a group incidence matrix $G_{i,g}$ is used to represent the count of each moiety as group $g$ for every metabolite $i$. We use the automated fragmentation algorithm and represent each group by a unique SMILES string utilizing the Cheminformatic tool RDkit[35] accessed through python.

### Bayesian ridge regression to determine the Gibbs energy of groups $\Delta_g G^o$

Using the molecular descriptions thus obtained, we determine the standard Gibbs energy contributions of moieties $\Delta_g G^o$ that allows the best fit of experimental data from TECRDB.

### Parameters

$S$: stoichiometric matrix ($R^{i \times j}$)
$G$: moiety incidence matrix that represents group decomposition ($R^{i \times g}$)
$\Delta_r G^o_{obs,j}$: Gibbs energy change of reaction j observed in enzyme thermodynamic database TECRDB.

### Variables

$\Delta_g G^o$: standard Gibbs energy contributions of moiety ($R^{g \times 1}$)
$\Delta_r G^o_{est}$: estimated standard Gibbs energy change of reactions ($R^{n \times 1}$)

The optimization formulation of multi-linear regression with L2 regularization (i.e., ridge regression) to estimate $\Delta_g G^o$ by minimizing the sum of the squared estimate of errors (SSE) and the regularization term is shown as follows:

$$\min \sum_j^n \left( \Delta_r G^o_{obs,j} - \Delta_r G^o_{est,j} \right)^2 + \lambda \left( \Delta_g G^o \right)^T \Delta_g G^o \tag{1}$$

s.t.

$$\Delta_r G^o_{est} = S^T G \cdot \Delta_g G^o \tag{2}$$

Because the multi-linear regression is prone to overfitting, especially since more moieties/

groups are defined in our method than previous GC-based methods, L2 regularization ($\lambda$ $(\Delta_g G^o)^T \Delta_g G^o$) is often applied in ridge regression to reduce cross-validation errors.

Confidence interval analysis from strongly biased models (e.g., ridge regression) relies on bootstrap-based methods. It often produces inaccurate uncertainty estimations [54]. Credible intervals used by Bayesian inference provide an alternative metric to confidence intervals. They have been shown to provide more reliable uncertainty estimation in 13C metabolic flux analysis [55]. Herein, we apply Bayesian ridge regression and define the prior of the $\Delta_g G^o$ as the isotropic Gaussian distributions with precision parameter $\alpha$:

$$p(\Delta_g G^o | \alpha) = N(\Delta_g G^o | 0, \alpha^{-1} I) \tag{3}$$

The likelihood function for the $\Delta_r G^o_{est}$ is defined as a Gaussian distribution with a precision parameter $\beta$:

$$p(\Delta_r G^o_{est} | \Delta_g G^o, \beta) = N(\Delta_r G^o_{est} | S^T G \cdot \Delta_g G^o, \beta^{-1}) \tag{4}$$

Based on the prior and likelihood function, $\Delta_g G^o$ is then estimated by the method of maximum a posteriori estimation (MAP) of the log of the posterior distribution:

$$\max_{\Delta_g G^o} ln(p(\Delta_r G^o_{est} | \Delta_g G^o, \beta) \cdot p(\Delta_g G^o | \alpha) \tag{5}$$

Note that the maximization of the log of the posterior distribution is equivalent to the multi-linear regression with L2 regularization defined in Eq (1) and $\lambda = \alpha/\beta$ as shown in [27]:

$$ln\left( p\left(\Delta_r G^o_{est} | \Delta_g G^o, \beta\right) \cdot p\left(\Delta_g G^o | \alpha\right) \right) = -\frac{\beta}{2} \sum_j^n (\Delta_r G^o_{obs,j} - \Delta_r G^o_{est,j})^2 - \frac{\alpha}{2} (\Delta_g G^o)^T \Delta_g G^o \tag{6}$$

The prediction interval of $\Delta_r G^o_{est,j}$ for a new reaction $j$ with a moiety change vector $x_j$ from Bayesian ridge regression can then be calculated as:

$$p(\Delta_r G^o_{est,j} | x_j, \Delta_g G^o, \alpha, \beta) = N(x_j \cdot \Delta_g G^o, \sigma^2(x_j)) \tag{7}$$

where the variance of the prediction interval can be calculated following the derivation in [34] as:

$$\sigma^2(x_j) = 1/\beta + x_j^T S_N x_j$$

$$S_N = (\alpha + \beta (S^T G)^T S^T G)^{-1}$$

We apply the Bayesian ridge regression function from the scikit-learn python package to train the Bayesian ridge regression model. Scikit-learn optimizes the precision parameters $\alpha$ and $\beta$ by iterative re-estimation based on an estimate for "how well-determined" the corresponding $\Delta_g G^o$ is by the training data [28]. We found the same parameters and mean squared error in the fitted data upon Bayesian ridge regression to 50 different initial guesses of precision parameters. We infer that the iterative method applied in Bayesian ridge regression in scikit-learn can produce unique, optimized parameters $\alpha$ and $\beta$. Then, the Bayesian ridge regression model can estimate the mean ($x_j \cdot \Delta_g G^o$) and standard deviation ($\sigma$) of $\Delta_r G^o_{est,j}$.

## Neural networks models for estimating Gibbs energy of groups $\Delta_g G^o$

We choose a feed-forward multi-layer perceptron neural network for nonlinear regression. These networks have neurons that are ordered in layers. The model starts with an input layer

(moiety incidence vector), followed by a hidden layer, and ends with an output layer. S1 Fig shows the architecture for a feed-forward neural network. The multi-layer perceptron model maps the moiety incidence matrix of the training data with the output layer using nonlinear functions. Finally, the output of the hidden layers is determined by using a rectified linear unit transfer function [56].

We applied the multi-layer perceptron regression from the scikit-learn python package to train the neural network model. LBFGS (Limited memory Broyden-Fletcher-Goldfarb-Shanno) backpropagation algorithm [29] is used to minimize the mean squared error and update the weights of the hidden layers based on the estimates in the output layer. The LBFGS is a faster technique in the family of quasi-newton methods commonly used for parameter estimation in machine learning [29]. Our model has three layers: input, hidden, and output. We considered the scikit-learn package default single hidden layer with 100 neurons to build the neural network model. Studies suggest that a single layer can adequately approximate any function which maps one finite space to another [30]. Scikit-learn package automatically estimates the number of neurons based on the cross-validation results for an accurate fit.

At last, we build three different models for bonding distance one, two and combining both distances. The M1,2-nonlinear model inputs the moiety incidence matrix for both the radiuses in a single input matrix. The radius one and two have 263 and 1,380 unique moieties for 673 metabolites in 4,001 reactions. Therefore, the M1,2-nonlinear model considers a total of 1,643 moieties to generate moiety incidence. The same leave-one-out cross-validation was performed as linear regression models for a direct comparison of the model performance.

## Pseudoisomers

The exact structure of the metabolite inside a cell is typically unknown because it exists as a mixture of pseudoisomers with different protonation states. Pseudoisomers are incorporated into dGPredictor by assuming that the intracellular mixture of pseudoisomers follows the Boltzmann distribution. We use Inverse Legendre Transform [12] to calculate the difference between the standard Gibbs energy of formation of a compound $\Delta_f G^o$ of the major pseduoisomer (i.e., most abundant at pH 7 and ionic concentration 0.25M) and the transformed Gibbs energy of formation of the mixture $\Delta_f G'^o$:

$$\Delta_f G'^o = \Delta_f G^o + RTln(10) \cdot m \cdot pH$$

$$- RTln\sum_{n=0}^{N_H}\exp\left[\ln(10)\sum_{i=m+1}^{n}(pKa(i) - pH) + \frac{2.91482(z_n^2 - n)\sqrt{I}}{RT(1 + 1.6\sqrt{I})}\right] \quad (8)$$

where the acid-base dissociation constant $pKa(i)$ of pseudoisomer $i$ is calculated using ChemAxon Marvin, $N_H$ is the maximum number of protonated hydrogens within a molecule (i.e., the number of $pKas$), $m$ is the number of hydrogens of the reference pseudoisomer, $z_n$ is the total charge of the pseudoisomer $n$, $I$ is the ionic strength, $R$ is the gas constant, and $T$ is the temperature.

The experimental measurement in TECRDB for a reaction for the apparent equilibrium constants $K'$, which is used to calculate the standard transformed Gibbs energy of reaction $j$:

$$\Delta_r G_j'^o = -RTln(K\prime) = \sum_i S_{ij}\Delta_f G'^o \quad (9)$$

Replacing $\Delta_f G'^o$ in Eq (9) with Eq (8), the resulting linear equations can be used to calculate $\Delta_f G^o$ using Gaussian elimination. When predicting $\Delta_r G'^o$ of a reaction, the difference between $\Delta_f G'^o$ and $\Delta_f G^o$ (i.e., $\Delta\Delta G = \Delta_f G'^o - \Delta_f G^o$) of all the metabolites in the reaction calculated from Eq (8) can be added to the $\Delta_r G^o$ calculated from regression analysis in the above sections to

correct the contribution of the pseudoisomer mixture. Both the calculation of $\Delta_f G^o$ using Inverse Legendre Transform and $\Delta\Delta G$ are implemented using the functions from the component contribution package (https://github.com/eladnoor/component-contribution).

## Supporting information

**S1 Table. The list of moieties generated for bonding distance one for all the TECRDB metabolites.**
(TXT)

**S2 Table. The list of moieties generated for bonding distance two for all the TECRDB metabolites.**
(TXT)

**S3 Table. The list of no group change reaction in training and validation dataset and MSE and mean MAE scores for Group Contribution method.**
(XLSX)

**S4 Table. The list of all reactions resolved by dGPredictor from the training dataset with no-group change and corresponding experimental and estimated Gibbs energies.**
(XLSX)

**S5 Table. Gibbs energy information of 3603 reaction rules and their corresponding KEGG IDs.**
(CSV)

**S6 Table. The list of KEGG reactions spanning unmeasured moieties**
(XLSX)

**S1 Fig. The architecture for multi-layer perceptron neural network.** An input layer of moiety incidence matrix ($j$ x $g$ matrix), a hidden layer with n neurons, and an output layer of estimates with j neurons. Here $j$ represents the reactions, g represents the groups/moieties, and n represents the neurons in the hidden layer.
(TIF)

## Acknowledgments

We thank the input given by Debolina Sarkar for helpful discussions.

## Author Contributions

**Conceptualization:** Lin Wang, Costas D. Maranas.

**Data curation:** Lin Wang, Vikas Upadhyay.

**Formal analysis:** Lin Wang, Vikas Upadhyay.

**Funding acquisition:** Costas D. Maranas.

**Methodology:** Lin Wang, Vikas Upadhyay.

**Supervision:** Costas D. Maranas.

**Validation:** Vikas Upadhyay.

**Visualization:** Lin Wang, Vikas Upadhyay.

**Writing – original draft:** Lin Wang, Vikas Upadhyay, Costas D. Maranas.

**Writing – review & editing:** Lin Wang, Vikas Upadhyay, Costas D. Maranas.

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
