## [Decision Letter · Decision Letter 0]

9 Apr 2021

Dear Dr. Maranas,

Thank you very much for submitting your manuscript "dGPredictor: Automated fragmentation method for metabolic reaction free energy prediction and de novo pathway design" for consideration at PLOS Computational Biology.

As with all papers reviewed by the journal, your manuscript was reviewed by members of the editorial board and by several independent reviewers. In light of the reviews (below this email), we would like to invite the resubmission of a significantly-revised version that takes into account the reviewers' comments.

We cannot make any decision about publication until we have seen the revised manuscript and your response to the reviewers' comments. Your revised manuscript is also likely to be sent to reviewers for further evaluation.

Sincerely,

Daniel A Beard

Deputy Editor

PLOS Computational Biology

Daniel Beard

Deputy Editor

PLOS Computational Biology

Reviewer's Responses to Questions

**Comments to the Authors:**

Reviewer #1: The paper by Wang et al. describes a new method for estimating Gibbs energy changes of chemical reactions. In comparison with previously developed Group/Component Contribution methods where molecules were decomposed into manually selected functional groups, Wang et al. describe an approach where they split a molecule into chemical moieties with distance up to 2 bonds. They performed regression analysis based on the experimental data from a thermodynamic database, obtaining a model that is able to predict Gibbs energy change of a reaction with high accuracy. This approach allows to include stereochemistry in the energy estimation, something that was not achieved with previously developed methods. Overall the paper describes a useful methods, is well written and I would recommend it for publication after several major and minor points are addressed.

Major points:

1. While the authours do provide a convenient browser-based tool to calculate the Gibbs energy of a reaction, they do not provide an intuitive Python tutorial to calculate Gibbs energies for a set of reactions. One clear jupyter notebook with the same example as shown in the Demo gif on github would suffice this purpose. This is a crucial step for the method to be used by the community.

2. Explanation related to Figure 4D at lines 326 - 339 is flawed. The authors explain that moiety C[C(@ or @@)H](O)(O) is not chiral in RDKit due to absence of explicit protons at C atom. This is wrong. The moiety is not chiral because in this SMILES notation it’s attached to two oxygen atoms. Since there are no branching after both oxygens the C atom in this moiety is not stereogenic. I do agree that if one looks for connectivity over two bonds (M2) then this moiety is chiral, but the explanation should be corrected. Additionally, RDkit does provide an option to draw hydrogen atoms explicitly (see https://www.rdkit.org/docs/GettingStartedInPython.html#modifying-molecules).

3. Authors do not mention anything about reactions including Fe or polymeric structure (starch, glycogen etc.). How well does their methods handle such compounds?

Minor points:

1. In the first paragraph/sentence of the introduction, the authors motivate their work, primarily from a metabolic engineering point of view. However, thermodynamics have been used in the context of metabolism in a much broader context. Thus, by placing their work in a bigger context, they could further increase the impact of their work.

2. In the 3rd sentence of the introduction, it sounds like beta-oxidation is an engineered pathway.

3. Could authors add a short comment regarding bias of predictions towards the compounds in the database. For example, I imagine there many more L-amino acids than D-amino acids in the database. How would it affect the predictions for D-amino acids?

4. Could the authors specify units in the equation at Line 285 and check if units for sigma (J/mol) are correct? I suspect it should be kJ/mol. Also provide units at Line 291.

5. One of the revisions of the Equilibrator included additional information on interaction with Mg2+. It’s known that this ion affects thermodynamics of many biochemical reactions due to binding. Do the authors consider influence of Mg?

6. I wonder if it’s possible to decompose the confidence interval of the Gibbs energy of a reaction into a standard deviation obtained from the regression and a standard deviation obtained for unknown moieties (33.29 (k?)J/mol).

7. The eQuilibrator website has turned into THE source for the community. I would strongly encourage that the authors of this paper here get in touch with the people behind the eQuilibrator to see whether the new code could be implemented into the eQuilibrator. Otherwise, I fear that the community, even though the data/tool developed here is superior to what is behind the eQuilibrator, will continue to use eQuilibrator.

8. In the discussion the authors mention that the new tool can now be used to predict for which reactions/compounds we would need measurements to further improve the predictions. I would greatly appreciate if the authors could add such analysis such that they could add some concrete suggestions on what we as a community should measure.

9. According to the IUPAC suggestions, the term “free” should be dropped in Gibbs FREE energy, see https://goldbook.iupac.org/terms/view/G02629. Thus, I suggest that the authors only use “Gibbs energy”.

10. In some parts of the manuscript the authors use GC (group contribution) when referring to the work of Noor et al, and later they use CC (component contribution). It would be great if the authors could make this consistent.

11. Line 187-188: I didn’t fully get this sentence.

12. Lines 218-22”" Is there an intuitive explanation for this finding? If there is, maybe the authors could hint to this?

13. Line 243-244: I didn’t get it. Also, I think the regularization appears here for the first time. It would be good if the authors could introduce this before; what do you put in the regularization term?

14. Comparison of lines 260 and 268: Here, I am confused (but maybe I miss a point). In line 260, it is stated that dGPredictor can cover all metabolites; while in line 268 it is stated that only 69% of allr reactions can be estimated. What is the reason that even though that all metabolites can be described we still cannot parametrize all reactions?

15. Lines 286-288: Please add some more explanation.

16. Line 329: Word missing?

17. Line 183: Gibb’s

18. Figure 4B is mentioned in the text before Figure 4A.

19. Line 386: Group change or moiety change?

Reviewer #2: In this work, Wang et al. present their automated method for metabolite decomposition and show how it can be applied within a framework similar to group contribution to predict standard GIbbs free energies. Using a standard function such as "FindAtomEnvironmentOfRadiusN" from RDkit is a very elegant solution and indeed expert-curated functional group definitions are no longer needed, group contribution-based methods would benefit greatly in terms of simplicity and generalizability.

The text is well-written, in a clear language, and the flow is logical and coherent.

I do, however, have major concerns regarding the analysis of the method and they way it was compared to previously published works on this topic:

* Line 192: "Model prediction accuracy" is defined as the MSE over the entire dataset. Since this is not a cross-validation error, I disagree that it represents prediction accuracy, but rather the goodness of fit of the model to the training data. Comparing this measure between algorithms that have very different degrees of freedom is quite misleading and should at least be corrected using the AKAIKE information criterion. In any case, the cross-validation prediction accuracy score is the *only* valid way to compare prediction performance and referring to it as "overfitting potential" seems to understate its real value. The fact that GC performs best among all tested methods is a very big caveat to this work and the authors must address this point and convince the readers how the other advantages of their method could counterweigh this disadvantage.

* Line 439: "MAE … was comparable to the current state-of-the-art CC method …". However, it seems that the comparison was done with the less advanced "group contribution" method. If this is indeed the case, why was the new method not compared to the state of the art (e.g. the method by Du et al. from 2018)? The authors should add this to the comparison table, or give reasons why this is not a fair comparison (and change the text to clearly say what was it compared to).

One of the discussed advantages of the proposed method is the ability to more accurately predict non-zero energies for isomerization reactions. Although it sounds reasonable, there is no real data to back this claim. We are only given one example (which could also just be a coincidence). A more thorough analysis of all such reactions, e.g. in the form of a cross-validated mean error, would be necessary to support the current claim.

* Although highlighted in the title of this paper, the thermodynamic tool for de novo pathway design and the text describing it are not well-developed. First of all, it's not very clear what the novelty of this tool is (besides the fact that it uses the dGPredictor library). In fact, more advanced methods for thermodynamic feasibility of designed pathways are available for many years (e.g. https://doi.org/10.1093/bioinformatics/bti213) and it seems that the current implementation is a step backward in terms of sophistication. The way irreversible reactions are defined is wrong (the standard ΔG'0 doesn't have to be both positive and negative within the confidence interval, but rather based on concentration ranges). The naive criterion of |ΔG'0| > 20 kJ/mol can quite often lead to mistakes, and the authors should consider replacing it with the established methods.

* Although I do see the potential in a tool such as the one shown in figure 5, which could be an interesting addition to the existing ones, I was not able to evaluate it due to technical reasons (see Minor point #2). That being said, even if the software would work, it's not clear what the scientific novelty of it would be. It seems more as a software integration issue and probably doesn't belong in this manuscript at all.

Minor points:

* When adding new compounds using InChI, how are the proton dissociation constants obtained for it? ChemAxon Marvin is mentioned once in the manuscript but it doesn't seem to be used anywhere in the scripts. Without these pKa values, any estimation of reactions with novel compounds would not be valid.

* In line 179: "to standardize the experimental values … at pH 7 and ionic strength 0.25 M". This is not correct, as the inverse Legendre Transform converts the biochemical Gibbs energies to "chemical" energies, where pH and ionic strength are undefined.

* I was not able to run the code stored on GitHub. Each of the Jupyter notebooks fails to run until the end due to missing files: "./data/median_b_extended.json" needed in the Jupyter notebook nalysis_dGPredictor.ipynb does not exist, "./model/bayesianRL_ac_all_model.sav" required for analyze_prior.ipynb is missing and so is "data/reaction_rule_r2_py3_manual_modified.csv" which is needed in new_linear_combine.ipynb. In addition, the Streamlit webapp didn't work, probably due to specific versions required for some of the dependencies (there is no requirements.txt or setup.cfg file as is standard in python packages). These technical issues should all be solved before the manuscript is published.

* The added value of figure 7 is not clear, as it shows a rather standard MLP network. Given the poor results, it seems unnecessary to dedicate a whole figure to it.

**Have all data underlying the figures and results presented in the manuscript been provided?**

Reviewer #1: None

PLOS authors have the option to publish the peer review history of their article (what does this mean?). If published, this will include your full peer review and any attached files.

Reviewer #1: No

Reviewer #2: No

**Have the authors made all data and (if applicable) computational code underlying the findings in their manuscript fully available?**

Reviewer #2: **No: **Some files are missing in the GitHub repository (see review comments)
---

## [Decision Letter · Decision Letter 1]

8 Jul 2021

Dear Dr. Maranas,

Thank you very much for submitting your manuscript "dGPredictor: Automated fragmentation method for metabolic reaction free energy prediction and de novo pathway design" for consideration at PLOS Computational Biology. As with all papers reviewed by the journal, your manuscript was reviewed by members of the editorial board and by several independent reviewers. In light of the reviews (below this email), we would like to invite the resubmission of a revised version that takes into account the reviewers' comments.

Both reviewers judge your contribution to important and potentially impactful. However, both reviewers still raise issues that need attention in order for your work to have the best possible impact. Please note that both reviewers report major difficulties in accessing, installing, and using your resource.

We cannot make any decision about publication until we have seen the revised manuscript and your response to the reviewers' comments. Your revised manuscript is also likely to be sent to reviewers for further evaluation.

Sincerely,

Daniel A Beard

Deputy Editor

PLOS Computational Biology

Daniel Beard

Deputy Editor

PLOS Computational Biology

Reviewer's Responses to Questions

**Comments to the Authors:**

Reviewer #1: The authors have addressed our minor comments sufficiently. Our main concerns remain: their code implementation (Major point #1 previously) and in part their reply to Major point #2.

Code (Major point 1):

1) The author’s choice to do their work in Python 2.7 is problematic because it is deprecated since 01.01.2020. Even some dependencies that the authors use (i.e. steamlit) are no longer supported, which makes the installation guide that they provided for the installation unusable. We could not execute it. This problem must be fixed.

2) Second, while the web interface might be suitable to look up values for a couple of reactions, we also wondered about a more convenient way to “bulk-retrieve” data for many reactions, as it might be required for genome-scale models.

Major point #2:

We think that the author’s explanation is still somewhat incomplete with regards to this point. The authors state: "… This is because the RDkit tool used for computing moieties stores molecules with implicit hydrogen (i.e., not explicitly present in the molecular graph) and *ignores the “C-H/O-H” bond when* generating the SMILES string. Thus, RDkit cannot identify the *chiral center when the “C-O-H” and “C-O-” bonds are responsible* for tetrahedral stereochemistry”. What is confusing is that the authors say "C-H, O-H are ignored", but then the authors mention the problem of a chiral center with C-O and C-O-H. In fact, chirality is also lost in SMILES notation for other bonds, e.g. C-C bonds, as in the example with 3-methyhexane CH3 - CH2 -CH (CH3) - CH2 - CH2 -CH3 (Smiles: CC(C)CCCC). The third C atom from the left is only chiral when you consider the full molecule. When you look at M1 and M2 distances as authors do, this atom is not chiral in SMILES notation.

Reviewer #2: I'm afraid that the answers that the authors provided to my comments are not always sufficient.

- Indeed, dGPredictor significantly increases the coverage compared to previous methods, but the abstract claims "has a higher prediction accuracy as compared to existing GC methods...". At least according to table 2, this is an incorrect statement (the cross-validation MAE is slightly larger).

Please change this statement to be correct (comparable cross-validation performance, better goodness-of-fit and coverage). I would even drop the claim for better goodness-of-fit since it only indicates that the system is over-trained.

- Regarding the comparison with Du et al., the fact that it is easier to compare to, doesn't justify using the older method. In fact, the 185 new curated reactions could highlight the advantage of dGPredictor even further (since the new algorithm has more degrees of freedom and therefore gain more from the new data). It is curious to me that the authors chose to ignore Du's method because it "increase[s] the median MAE of the GC method", since that increase was very small (and comparable to the increase found in this study as well).

- The data regarding reactions with no group change seems quite strange. GC should always predict a zero reaction energy change (maybe slightly shifted do to the pKas), and therefore the MSE should simply be the variance of the ΔG0 data for these reactions. This should also be the case when doing cross-validation (since we don't actually use the linear model for this prediction). If this is the case, how can the MSE be 63.97 kJ/mol (much higher than the mean across all reactions in general - which is by itself not very likely) while the cross-validation MAE is only 3.84 kJ/mol (much lower than the average)?

In addition, the abstract claims that the proposed method is an improvement because it adds consideration of stereochemistry within metabolite structures. However, this claim is not tested directly, but only together with the use of the graph-based groups (instead of user defined ones). If stereochemistry is important - it can also be added to the user-defined groups. Perhaps doing that would create a better result than dGPredictor?

- The definitions and names in the manuscript are not very clear. I understand that dGPredictor is a catchy name for the code package, but it probably should be separated from the method itself. Using NovoStoich to replace the user-defined groups in GC is the main novelty and the name dGPredictor does not reflect that (i.e. it is too generic). In addition, the paper seems to throw in other tools that fit in the dGPredictor software suite, but are not very relevant to this work. I pointed out before that the tool for testing the reversibility of reactions is an inferior algorithm to other existing tools (I'm not going to repeat what I wrote here), and it seems that the only reason to mention it is its reliance on the dGPredictor library. It is my hope (and apparently also that of the other reviewer) that dGPredictor could be integrated into other existing software packages, so that its advantages would be easily accessible to the community. If the authors insist on keeping the dual focus (both on the algorithm itself, and on the software package), then they must improve the coding standards (see my last comment).

- A lot of the values in the text are missing units (mostly kJ/mol or kJ^2/mol^2). Here are a two out of the many other cases: Line 203, Table 2.

It's not clear why the fit is described by a Mean squared error, while the cross-validation is using Mean absolute error (although in figure 2 the median is shown with the same values, is it Mean or Median??). RMSE (Root Mean Squared Error) is the most common measure and can be applied in both cases (it was used to compare the last 6 predictors in the Du et al. review paper - DOI:10.1016/j.tibs.2018.09.009). In general, the cross-validation error should be higher than the fitting error and it's hard to see it when the metrics are different (also, see my comment about the no-group-change reactions).

Can the authors explain why GC has by far the worst MSE but also the best cross-validation MAE? Is it an indication that all other methods are more susceptible to over-training?

- Please improve the coding standards of your software package. Python 2.7 is unsupported since January 2020 (https://www.python.org/doc/sunset-python-2/). There are countless tools that can help one port the code to Python 3. Downloading a .zip file from a sharepoint folder cannot be part of the installation process - and completely defeats the purpose of hosting code on GitHub (it seems to be only relevant for the comparative analysis, so probably optional - it would be best to mention that and make sure it's not necessary). In addition, the README still has some errors. For example the first step is: Generate model file by "running model_gen.py" using "python model_gen.py". However, the file model_gen.py is not in the repository. The next step (running streamlit) did not work on my computer either. It could have to do with different platforms, but I urge the authors to try and facilitate running their tool on more platforms.

**Have the authors made all data and (if applicable) computational code underlying the findings in their manuscript fully available?**

Reviewer #1: **No: **see comment to the authors

Reviewer #2: Yes

PLOS authors have the option to publish the peer review history of their article (what does this mean?). If published, this will include your full peer review and any attached files.

Reviewer #1: No

Reviewer #2: No
---

## [Decision Letter · Decision Letter 2]

13 Sep 2021

Dear Dr. Maranas,

We are pleased to inform you that your manuscript 'dGPredictor: Automated fragmentation method for metabolic reaction free energy prediction and de novo pathway design' has been provisionally accepted for publication in PLOS Computational Biology.

Best regards,

Daniel A Beard

Deputy Editor

PLOS Computational Biology

Daniel Beard

Deputy Editor

PLOS Computational Biology

Reviewer's Responses to Questions

**Comments to the Authors:**

Reviewer #1: The authors have addressed my all my points.

Reviewer #2: The authors have addressed the most important comment regarding the software itself, namely porting it to Python v3 and making it relatively straightforward to install and use (on multiple platforms).

**Have the authors made all data and (if applicable) computational code underlying the findings in their manuscript fully available?**

Reviewer #1: Yes

Reviewer #2: Yes

PLOS authors have the option to publish the peer review history of their article (what does this mean?). If published, this will include your full peer review and any attached files.

Reviewer #1: No

Reviewer #2: No

---

## [Editor Report · Acceptance letter]

20 Sep 2021

PCOMPBIOL-D-21-00509R2 

dGPredictor: Automated fragmentation method for metabolic reaction free energy prediction and de novo pathway design

Dear Dr Maranas,

I am pleased to inform you that your manuscript has been formally accepted for publication in PLOS Computational Biology. Your manuscript is now with our production department and you will be notified of the publication date in due course.

With kind regards,

Katalin Szabo
